# Hemodynamic simulation in the aortic arch under anemic, diabetic, and healthy blood flow conditions using computational fluid dynamics

Farzana Akter Tina[1], Hashnayne Ahmed[1,2]*, Hena Rani Biswas[1]

1 Department of Mathematics, Faculty of Science and Engineering, University of Barishal, Barishal, Bangladesh, 2 Department of Mechanical and Aerospace Engineering, University of Florida, Gainesville, Florida, United States of America

* hashnayneahmed17@gmail.com

## Abstract

This study examines the hemodynamic impact of anemic, diabetic, and healthy blood conditions in the human aortic arch using computational modeling. Blood rheology was represented by the shear-thinning Carreau–Yasuda model, and simulations were carried out in a patient-inspired aortic geometry under pulsatile flow. Velocity fields, pressure gradients, and wall shear stress (WSS) distributions were quantified to assess how altered hematocrit and viscosity affect vascular loading. Anemic blood, characterized by low viscosity, showed smooth low-resistance flow with reduced WSS, potentially limiting endothelial stimulation and impairing perfusion. Diabetic blood exhibited elevated viscosity and hematocrit, producing higher flow resistance, increased WSS, and disturbed secondary flows, consistent with vascular stiffening and remodeling risk. Healthy cases maintained balanced hemodynamics within physiological ranges. These findings highlight the mechanistic links between blood rheology and vascular stress, offering non-invasive insights for risk stratification in hematological and metabolic disorders, and supporting the integration of CFD-based analysis into clinical decision-making.

## Introduction

Understanding blood flow dynamics in the aortic arch is essential for advancing cardiovascular diagnostics, risk stratification, and therapeutic planning. As a critical conduit for oxygenated blood to the brain and upper extremities, the aortic arch exhibits complex hemodynamic behavior driven by its curved geometry, branching vessels, and pulsatile flow. These features generate swirling streamlines, recirculation zones, and spatial gradients in wall shear stress, all of which are mechanistically linked to pathologies such as atherosclerosis, aneurysms, and hypertension [1–3]. Conventional imaging techniques often fall short in resolving these dynamic flow structures,

**Data availability statement:** All relevant data are within the manuscript.

**Funding:** The author(s) received no specific funding for this work.

**Competing interests:** The authors have declared that no competing interests exist.

whereas computational fluid dynamics offers a high-fidelity approach to simulate and analyze vascular hemodynamics in detail.

A growing body of literature has applied CFD to study aortic flow, demonstrating how vessel geometry and blood rheology shape hemodynamic forces. For instance, Numata et al. [3] showed that aortic dilation amplifies helical flow and increases the oscillatory shear index (OSI), revealing cerebroprotective effects from regional perfusion strategies. Prahl Wittberg et al. [4] highlighted the influence of vessel anomalies and red blood cell (RBC) dynamics on WSS distribution. Miyazaki et al. [5] validated turbulence models against 4D Flow MRI, while Markl et al. [6] emphasized the clinical importance of patient-specific CFD for visualizing flow asymmetries and identifying recirculation zones. Ahmed et al. [7] explored the effect of disease conditions such as anemia and diabetes on hemodynamic parameters, showing how pathological rheology can perturb flow uniformity and increase mechanical stress.

Building on these efforts, subsequent studies have incorporated multiphysics effects and non-Newtonian rheology to improve physiological accuracy. Caballero and Laín [8] compared Newtonian and Carreau–Yasuda models, concluding that shear-thinning behavior is critical for resolving near-wall flow. Silva et al. [9] investigated fluid–structure interaction (FSI) in post-endograft flows, showing how device implantation alters WSS and introduces turbulence. Additional studies across carotid and coronary arteries have associated low WSS and oscillatory flow with plaque formation and arterial remodeling [2,10]. Yet, despite these advances, few studies have systematically compared multiple blood conditions such as anemic, diabetic, and healthy states within a unified CFD framework. This absence of comparative rheological insight in the aortic arch represents a key gap in current literature.

Recent investigations have also emphasized the role of geometric complexity, turbulence modeling, and experimental validation. Xu et al. [11] found that Large Eddy Simulation (LES) significantly improves WSS predictions in disturbed regions compared to laminar models. Zandvakili et al. [12] and Liu et al. [13] illustrated how bifurcation geometry shapes vortical structures and local shear gradients. Experimental studies by Zimmermann et al. [14] and Šeta et al. [15] further confirmed the presence of complex secondary flows and pressure fluctuations in the arch. While these efforts have improved realism, they often focus on either anatomy or turbulence but not on the comparative influence of pathological blood rheology, especially in the context of pulsatile aortic flow.

In this study, we address this gap by conducting a rheology-aware CFD analysis of blood flow in the aortic arch, comparing anemic, diabetic, and healthy conditions within a consistent simulation framework. A Carreau–Yasuda viscosity model is employed as a unified constitutive description to enable controlled comparison across blood conditions, and a pulsatile inlet condition is applied to capture physiologic flow dynamics. The simulation resolves transient velocity profiles, pressure distributions, and wall shear stress patterns, to assess how hematocrit- and viscosity-driven rheological variations influence mechanical stress and flow organization. By systematically isolating blood rheology as the primary varying factor, and coupling each outlet to a physiologically realistic three-element Windkessel (RCR) model, the present

framework enables mechanistic interpretation of pressure–velocity–WSS coupling in a central arterial conduit. This study provides a controlled, system-level assessment of how pathological blood rheology alone reshapes pressure, velocity, and shear organization in the aortic arch, revealing mechanistic effects that are not accessible from branch-level or single-condition analyses.

The remainder of this article is organized as follows: Sect 1 describes the physical and mathematical modeling framework, including governing equations, rheology, and boundary conditions. Sect 2 outlines the numerical approach, including geometry reconstruction, meshing, and solver configuration. Sect 3 presents the simulation results and interprets key trends in velocity, pressure, and WSS across blood conditions. Finally, Sect 4 summarizes the major findings, limitations, and implications for future clinical and computational research.

## 1 Physical and mathematical framework

The aortic arch is one of the most complex and clinically significant regions of the human arterial system, where the combination of curvature, bifurcations, and pulsatile blood flow gives rise to rich and varied hemodynamic behavior. Modeling such flow requires a geometry that faithfully reflects anatomical features, as well as a mathematical formulation capable of capturing unsteady, three-dimensional, and non-Newtonian effects. In this study, we focus on quantifying physiologically relevant flow structures particularly wall shear stress under pulsatile conditions within a realistic aortic arch geometry.

To this end, the anatomical structure of the aortic arch is reconstructed from medical imaging data, capturing its major branches: the brachiocephalic, left common carotid, and left subclavian arteries. These branches are associated with strong geometric asymmetry and abrupt changes in curvature, both of which significantly influence the local distribution of shear forces and secondary flows. The geometry used in this study was obtained from a publicly available CAD model (https://grabcad.com/library/human-aortic-arch-1) and refined to eliminate non-physiological surface irregularities. Fig 1 shows the MRI scan and schematic diagram used to guide the modeling process. The geometry preserves critical branching angles and curvature radii that influence local hemodynamics.

From a physical modeling standpoint, the dynamics of blood flow in the aortic arch are governed by unsteady, incompressible fluid mechanics and by the non-Newtonian rheological behavior of blood. Because of the pulsatile nature of cardiac output and the geometric complexity of the domain, a time-dependent formulation of the Navier–Stokes equations is required. The governing equations for mass and momentum conservation are given in Cartesian co-ordinates as follows:

$$\nabla \cdot \mathbf{u} = 0 \tag{1}$$

$$\rho \frac{\partial \mathbf{u}}{\partial t} + \rho(\mathbf{u} \cdot \nabla)\mathbf{u} = -\nabla p + \nabla \cdot \tau \tag{2}$$

where $\mathbf{u}$ is the velocity field, $p$ is pressure, $\rho$ is the density of blood, and $\tau = \mu(\dot{\gamma})\left(\nabla \mathbf{u} + \nabla \mathbf{u}^T\right)$ is the extra stress tensor with shear-rate dependent viscosity.

In physiological conditions, blood density typically ranges from 1050 to 1060 kg/m$^3$, while apparent viscosity varies from 0.003 to 0.16 Pa · s, depending on the local shear rate. These properties play a critical role in modulating both inertia and wall shear. The equations account for the inertial, pressure, and viscous forces acting on the fluid, with the unsteady term capturing the time-varying nature of cardiac output. To incorporate the shear-thinning behavior of blood, the dynamic viscosity $\mu$ is defined using the Carreau–Yasuda model:

$$\mu_{\text{eff}}(\dot{\gamma}) = \mu_\infty + (\mu_0 - \mu_\infty)\left(1 + (\lambda\dot{\gamma})^2\right)^{\frac{k-1}{2}} \tag{3}$$

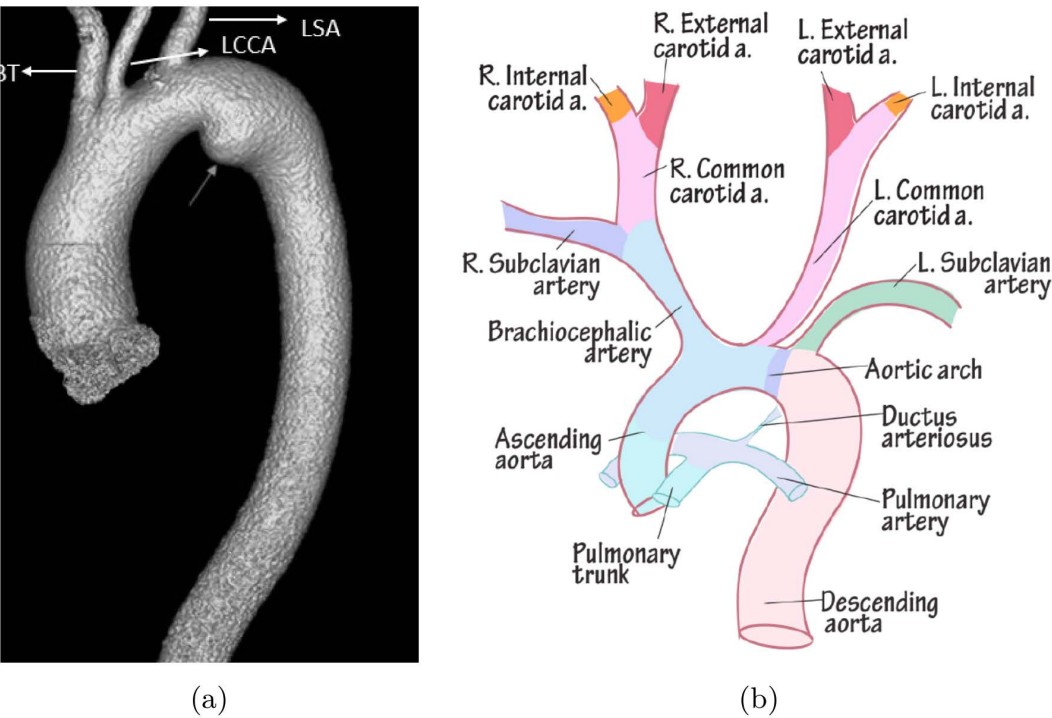

**Fig 1. Visualization of the aortic arch. (a)** MRI scan showing the anatomical structure of the aortic arch. Adapted from Baz et al. [16]. **(b)** Schematic representation of the aortic arch highlighting key anatomical branches. Adapted from Natiq et al. [17].

where $\dot{\gamma}$ is the local shear rate, $\mu_0$ and $\mu_\infty$ are the viscosities at zero and infinite shear rate, respectively, $\lambda$ is a time constant, and $k$ is the power-law index. This model effectively captures blood's shear-thinning behavior through an effective viscosity formulation, particularly in the near-wall and low-flow regions common in curved arterial segments.

Boundary conditions are prescribed to approximate in vivo hemodynamics. At the inlet of the ascending aorta, a pulsatile velocity waveform is applied to simulate periodic cardiac input. The waveform, based on the model by Owasit et al. [7,18], is given by:

$$v_{\text{inlet}}(t) = \begin{cases} 0.5 \sin\left[4\pi\left(t + 0.016\right)\right], & \text{if} \quad 0.5n < t \leq 0.5n + 0.218 \\ 0.1, & \text{if} \quad 0.5n + 0.218 < t \leq 0.5(n + 1) \end{cases} \tag{4}$$

where $n = 0, 1, 2, \ldots$, and the waveform spans a cardiac cycle duration of 0.5 seconds (corresponding to 120 beats per minute). This representation captures the acceleration and deceleration phases of systole and diastole. The Reynolds number based on peak systolic velocity and vessel diameter falls in the transitional regime ($\sim 1000$–$2000$), indicating the potential for complex vortical structures.

The aortic arch model comprises four outlets: the brachiocephalic artery ($O_1$), left common carotid artery ($O_2$), left subclavian artery ($O_3$), and descending aorta ($O_4$). Each outlet was coupled with a three-element Windkessel (RCR) model to represent downstream vascular resistance and compliance [19–21]. This model provides a physiologically realistic pressure–flow coupling by accounting for both the resistive and compliant effects of the peripheral vasculature, allowing outlet pressure to evolve dynamically with time rather than remain fixed. Each branch was assigned parameters ($R_{1,i}, R_{2,i}, C_i$), denoting proximal resistance, distal resistance, and compliance, respectively, and related through the Windkessel

formulation $P_i(t) = P_{c,i}(t) + R_{1,i}Q_i(t)$, where $P_{c,i}$ varies with capacitive storage and distal resistance. The governing relation was discretized using a first-order backward-Euler scheme to ensure numerical stability:

$$P_i^{n+1} = \frac{\beta_i P_i^n + Q_i^{n+1}\left(R_{2,i} + R_{1,i} + R_{1,i}\beta_i\right) - R_{1,i}\beta_i Q_i^n}{1 + \beta_i}, \qquad \beta_i = \frac{R_{2,i}C_i}{\Delta t}. \tag{5}$$

The updated pressure $P_i^{n+1}$ was applied at each time step as the outlet boundary condition, allowing physiological wave reflections and pressure decay throughout the cardiac cycle. The total resistance for each branch was estimated as $R_{\text{tot},i} = \bar{P}_{\text{mean}}/\bar{Q}_{i,\text{mean}}$, with $R_{1,i} = 0.05\,R_{\text{tot},i}$, $R_{2,i} = 0.95\,R_{\text{tot},i}$, and $C_i = \tau/R_{2,i}$, where $\tau \approx 1.0\,\text{s}$ is the diastolic decay constant. The mean flow partitions among the four outlets followed the ratios reported by Reymond *et al.* [21].

Vessel walls are assumed to be rigid with a no-slip boundary condition $\mathbf{u} = 0$. While this approximation neglects wall compliance, it simplifies the problem and isolates the effect of flow pulsatility and geometric features. Future work may incorporate fluid–structure interaction (FSI) to assess the influence of wall motion.

To quantify near-wall hemodynamics, the wall shear stress $\tau_{\text{wss}}$ is the tangential force per unit area exerted by the fluid on the arterial wall, given by $-\mu_{\text{eff}}\left(\partial u_t/\partial \eta\right)_{\text{wall}}$, where $\mu_{\text{eff}}$ is the effective viscosity, $u_t$ is the tangential velocity, and $\eta$ is the local surface-normal direction. For nondimensional interpretation, the skin friction coefficient, $C_f$ is calculated as $\tau_{\text{wss}}/(0.5\rho v^2)$, where $\rho$ is the fluid density. To further assess the rotational complexity of blood flow in the aortic arch, we computed the helicity field, a scalar measure of the alignment between velocity and vorticity vectors [22]. Helicity captures the presence of swirling and helical flow structures, which are prominent in curved and branching vessels and defined as,

$$\mathcal{H} = u\left(\frac{\partial w}{\partial y} - \frac{\partial v}{\partial z}\right) + v\left(\frac{\partial u}{\partial z} - \frac{\partial w}{\partial x}\right) + w\left(\frac{\partial v}{\partial x} - \frac{\partial u}{\partial y}\right), \tag{6}$$

where $u, v, w$ are the velocity components in the $x, y, z$ directions, respectively. These quantities serve as critical indicators of endothelial shear exposure and are used throughout this study to identify regions of disturbed flow, shear imbalance, and vascular risk. Together, the physical and mathematical framework presented here enables a high-fidelity investigation of pulsatile, three-dimensional, non-Newtonian blood flow in the aortic arch. This formulation enables investigation of flow disturbances near branching points, which are often associated with early-stage atherosclerotic development.

## 2 Numerical methodology

To perform physiologically accurate simulations of blood flow in the aortic arch, the computational domain was constructed from a patient-inspired aortic arch geometry derived from publicly available MRI-based anatomical reconstructions, rather than from subject-specific clinical imaging data and converted into a surface mesh in STL format. This file was preprocessed using CAD-based software to ensure anatomical consistency and compatibility with the meshing pipeline. Preprocessing operations such as smoothing and artifact removal were necessary to eliminate abrupt surface irregularities that could lead to numerical instability. This step ensures the final model remains both anatomically faithful and numerically robust. Fig 2 illustrates the reconstructed geometry and its discretized domain.

To verify that flow predictions were independent of spatial discretization, a grid sensitivity analysis was conducted. This test was crucial to ensure that numerical results, particularly velocity and wall shear stress (WSS), remained consistent as the mesh was refined. Several unstructured tetrahedral meshes were tested by varying the maximum element size from 4.0 mm to 0.7 mm, while holding all physical models and boundary conditions constant, in order to assess convergence of outlet velocity and wall shear stress predictions. As shown in Table 1, changes in key hemodynamic quantities diminished beyond a mesh size of 1.0 mm, with further refinement producing negligible variation, confirming convergence. Accordingly, the selected mesh represents an optimal compromise between numerical accuracy and computational cost, enabling accurate yet efficient simulations.

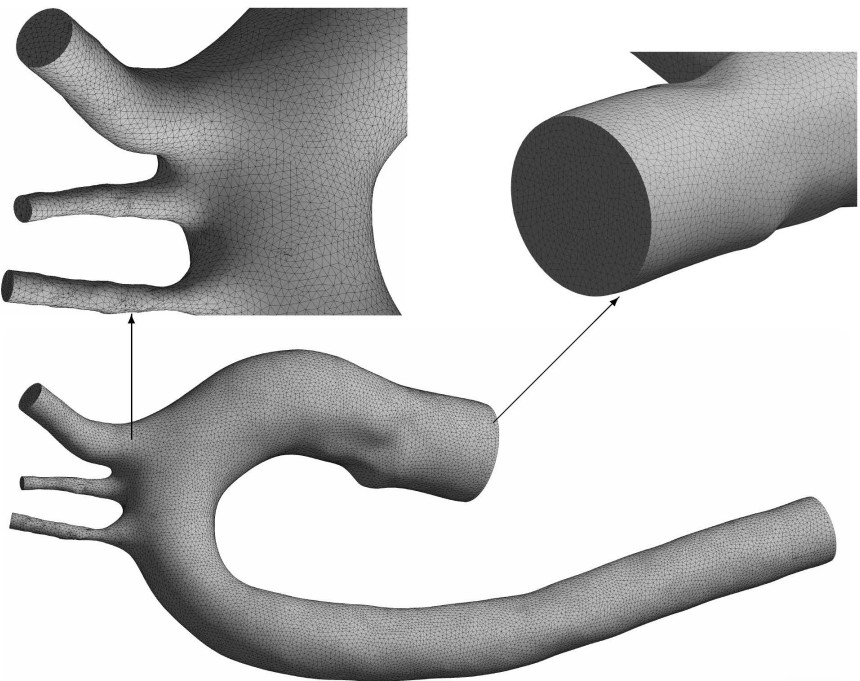

**Fig 2. Unstructured tetrahedral mesh of the aortic arch.** Enlarged views on the left highlight local mesh refinement at a supra-aortic branch junction, while the view on the right illustrates near-wall resolution at the inlet.

**Table 1. Grid independence study showing the average velocity at four outlets and WSS under increasing mesh resolutions.**

| Elements | Max Size (mm) | Average Velocity (m/s) | | | | WSS (Pa) | |
|---|---|---|---|---|---|---|---|
| | | O1 | O2 | O3 | O4 | Average | Maximum |
| 48260 | 4.0 | 0.1680 | 0.1821 | 0.0953 | 0.0839 | 0.5941 | 4.7700 |
| 72526 | 3.0 | 0.1708 | 0.1808 | 0.0916 | 0.0847 | 0.5731 | 4.6384 |
| 98907 | 2.0 | 0.1539 | 0.1929 | 0.0909 | 0.0868 | 0.5382 | 3.6551 |
| 341047 | 1.0 | 0.1547 | 0.2053 | 0.0989 | 0.0875 | 0.4950 | 3.9756 |
| 417613 | 0.9 | 0.1523 | 0.2051 | 0.1046 | 0.0859 | 0.4870 | 3.6007 |
| 527893 | 0.8 | 0.1502 | 0.1987 | 0.1044 | 0.0879 | 0.4841 | 2.9467 |
| 690709 | 0.7 | 0.1543 | 0.1947 | 0.1072 | 0.0879 | 0.4763 | 3.0584 |

Based on this evaluation, a final mesh was adopted comprising 341,047 unstructured tetrahedral elements and 502,590 nodes. Local refinement was applied in regions of high curvature and near the origins of the supra-aortic branches to resolve steep gradients in velocity and shear. Additionally, boundary layer elements were introduced near the vessel walls. This step is essential for accurately capturing wall shear stress, a key hemodynamic parameter analyzed in this study.

The simulations were performed using a transient incompressible flow solver, ANSYS Fluent (https://www.ansys.com/). A pressure-based solver was selected, with the pressure–velocity coupling handled by the coupled scheme. This approach improves numerical stability, especially in complex geometries under unsteady conditions. Spatial derivatives were computed using the least-square cell-based method, and second-order upwind schemes were applied to the momentum equations. These higher-order methods reduce numerical diffusion and enhance accuracy in capturing

convective transport. A hybrid initialization strategy was adopted to generate the initial flow field, combining patch and standard techniques to accelerate convergence. Residuals for continuity and momentum equations were required to drop below $10^{-6}$ to ensure solution accuracy, with a maximum of 1000 iterations allowed per time step.

To validate the temporal resolution, a time-step sensitivity analysis was carried out. This analysis ensures that the temporal discretization does not distort the time-dependent solution. Six time steps were tested under constant total simulation duration, and their effects on peak velocity and WSS were evaluated. As shown in Table 2, the variation across step sizes was negligible, confirming that a time step of $0.4\,s$ is sufficient for resolving the pulsatile dynamics while keeping computational effort reasonable.

The Carreau–Yasuda model was employed to represent the non-Newtonian behavior of blood, capturing its shear-thinning response under varying flow conditions. A constant density of $1060\,kg/m^3$ was used for all cases. Simulations were carried out for four distinct blood conditions: anemic, diabetic, and two healthy profiles representing different rheological states. These conditions were selected to assess the effects of hematocrit variation and pathological viscosity changes on flow behavior. The specific parameters used in the model are summarized in Table 3.

Fig 3 illustrates the variation of effective blood viscosity with shear rate for the four blood conditions computed using the Carreau–Yasuda formulation in Eq (3). In all cases, viscosity decreases monotonically with increasing shear rate, demonstrating the characteristic shear-thinning behavior of blood. Diabetic blood exhibits the highest viscosity across the entire shear-rate range due to elevated zero-shear viscosity and a delayed transition to the high-shear plateau, whereas anemic blood shows the lowest viscosity in the moderate-to-high shear regime relevant to large-artery flow. The two healthy blood cases fall between these extremes, indicating moderate shear-thinning behavior and inter-individual rheological variability. Although the difference between the two healthy subjects is more pronounced in the very low-shear range ($< 1\,s^{-1}$), the viscosity deviations from Healthy Case 1 become comparable in magnitude for both Healthy Case 2 and the anemic case within the arterial shear range ($\mathcal{O}(10^2\text{–}10^3)\,s^{-1}$), consistent with shear rates estimated from reported mean arterial wall shear stresses (approximately 1–2 Pa) in large arteries [24,25]. Through Eq (3), these shear-dependent viscosity

**Table 2. Time-step sensitivity analysis for a maximum mesh size of** $1.0\,mm$**. The total simulation time was maintained at approximately** $10\,s$ **across six different time step sizes.**

| Time Step (s) | Total Steps | Maximum Velocity (m/s) | | | | WSS (Pa) | |
|---|---|---|---|---|---|---|---|
| | | Outlet 1 | Outlet 2 | Outlet 3 | Outlet 4 | Average | Maximum |
| 0.6 | 17 | 0.2450 | 0.2944 | 0.2292 | 0.1753 | 0.4950 | 3.9756 |
| 0.5 | 25 | 0.2450 | 0.2943 | 0.2292 | 0.1753 | 0.4950 | 3.9756 |
| 0.4 | 25 | 0.2450 | 0.2944 | 0.2292 | 0.1753 | 0.4950 | 3.9756 |
| 0.3 | 33 | 0.2450 | 0.2943 | 0.2292 | 0.1753 | 0.4950 | 3.9756 |
| 0.2 | 50 | 0.2450 | 0.2943 | 0.2292 | 0.1753 | 0.4950 | 3.9756 |
| 0.1 | 100 | 0.2450 | 0.2943 | 0.2292 | 0.1753 | 0.4950 | 3.9756 |

**Table 3. Rheological parameters used in the Carreau–Yasuda model to represent blood conditions, adapted from [7,23]. These values capture the shear-thinning behavior of blood under different physiological states.**

| Properties | Anemic | Diabetic | Healthy Blood | |
|---|---|---|---|---|
| | Blood | Blood | (Case 1) | (Case 2) |
| Power-law index, $k$ | 0.33 | 0.39 | 0.48 | 0.3568 |
| Constant time, $\lambda$ (s) | 12.448 | 103.09 | 39.418 | 3.313 |
| $\mu_\infty$ (Pa · s) | 0.00257 | 0.00802 | 0.00345 | 0.0035 |
| $\mu_0$ (Pa · s) | 0.0178 | 0.8592 | 0.0161 | 0.056 |
| Hematocrit count | 25% | 65% | About 45% | About 45% |

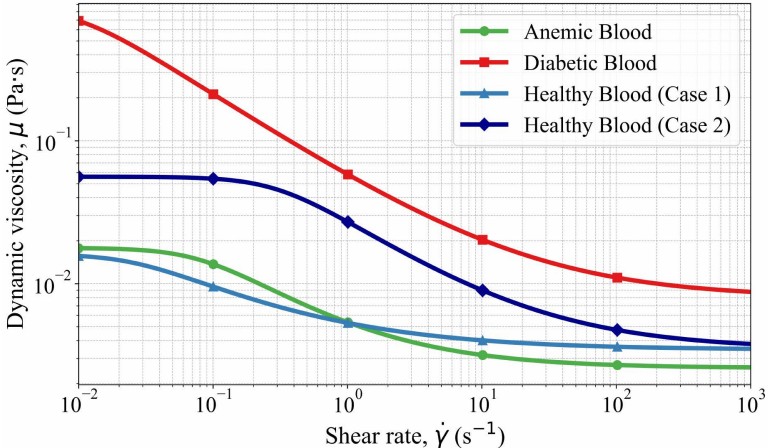

**Fig 3. Variation of effective blood viscosity with shear rate for four blood conditions based on the Carreau–Yasuda model defined in** Eq (3).

variations directly modulate the viscous stress term in the momentum equations, thereby influencing local velocity gradients and wall shear stress distributions throughout the pulsatile arterial flow field. Based on the computed wall shear stresses and corresponding effective viscosity values, the estimated wall shear rates are on the order of several hundred $s^{-1}$, confirming that the present aortic flow operates predominantly within this moderate-to-high shear regime characteristic of large arteries [24,25].

Each simulation was performed over a total physical duration of $10\,s$, covering multiple cardiac cycles and allowing fully developed pulsatile flow to be resolved. Together with the validated spatial and temporal resolutions and physiologically representative blood properties, this setup provides a robust numerical framework for investigating hemodynamic phenomena in the aortic arch.

## 3 Results and discussions

This section presents the simulation outcomes of blood flow in the aortic arch under four physiological and pathological conditions: anemic, diabetic, and two healthy cases. The results are analyzed through velocity distributions, pressure fields, and wall shear stress patterns to assess how blood rheology and geometry influence hemodynamic behavior. Each subsection focuses on a specific parameter, supported by both qualitative visualizations and quantitative comparisons across cases. These parameters are essential for understanding not only mechanical aspects of blood transport but also the onset of vascular dysfunctions such as atherosclerosis, hypertension, and thrombus formation. Integrating rheological properties with anatomical geometry helps bridge computational hemodynamics with clinical risk assessment [26,27].

### 3.1 Velocity profiles and flow topology

Velocity distribution and flow topology are key determinants of hemodynamic performance in the aortic arch. Figs 4 and 5 compare velocity streamlines and vector fields, respectively, across four blood conditions: anemic, diabetic, and two healthy cases. These visualizations highlight how blood viscosity, hematocrit level, and the arterial shape of the aortic arch influence overall flow characteristics, including secondary flow formation and recirculation. While streamlines emphasize bulk flow paths and recirculation, vector fields resolve local velocity magnitude and direction. Such flow patterns are physiologically significant because they modulate endothelial shear stress, particle residence time, and the likelihood of disturbed flow; all known to influence vascular health [27,28]. This is particularly relevant in curved arterial segments like

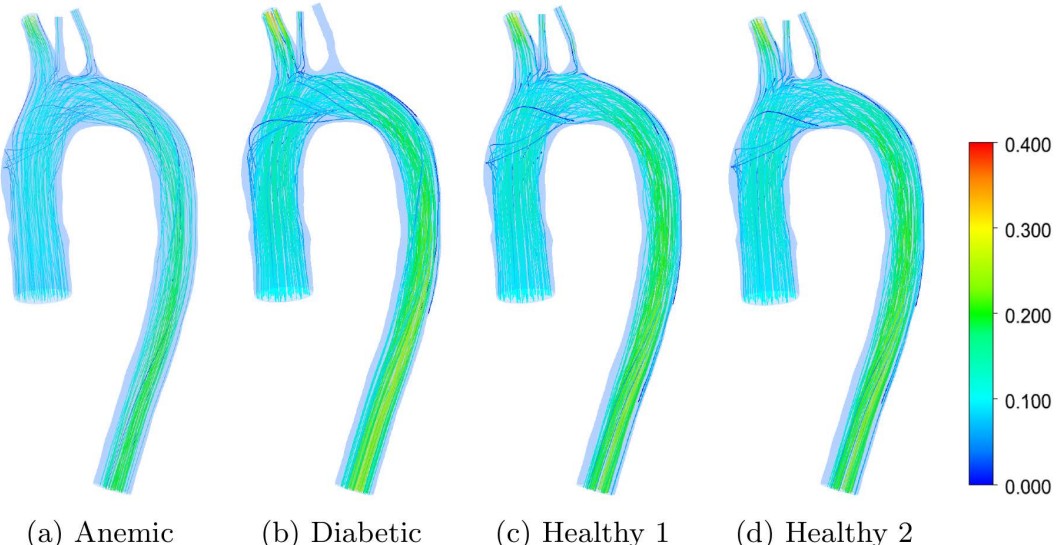

(a) Anemic (b) Diabetic (c) Healthy 1 (d) Healthy 2

**Fig 4. Velocity streamlines in the aortic arch for different blood flow cases: (a) anemic, (b) diabetic, (c) healthy (Case 1), and (d) healthy (Case 2).** Red regions represent higher velocity magnitudes; blue regions indicate lower values, in ms$^{-1}$.

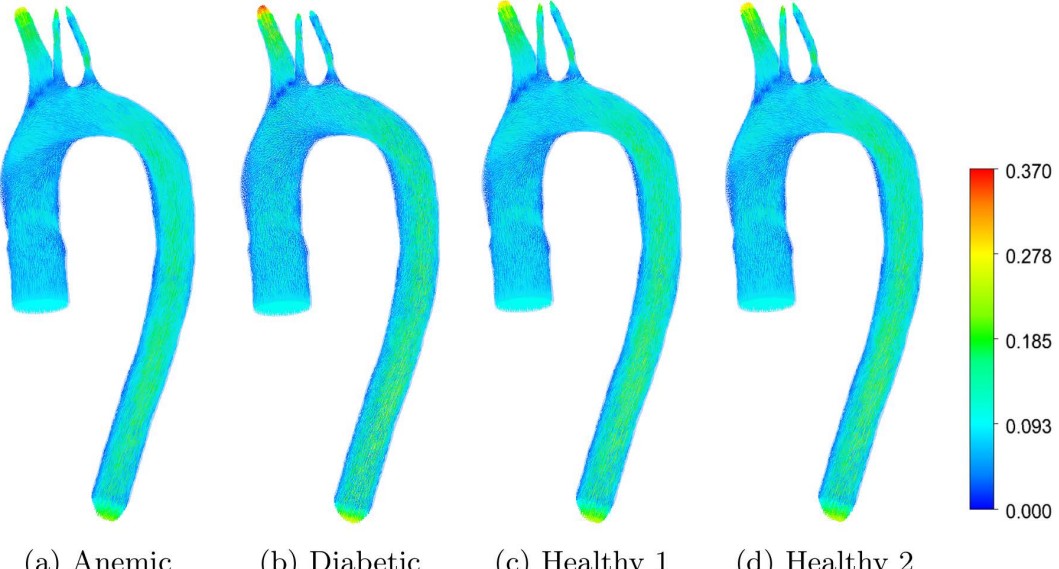

(a) Anemic (b) Diabetic (c) Healthy 1 (d) Healthy 2

**Fig 5. Velocity vector fields in the aortic arch for different blood flow cases: (a) anemic, (b) diabetic, (c) healthy (Case 1), and (d) healthy (Case 2).** Vector directions and magnitudes illustrate localized flow variations across conditions, in ms$^{-1}$.

the aortic arch, where centrifugal forces and secondary flow structures such as Dean vortices can amplify or suppress local disturbances, affecting both mechanical stress and particle transport [29].

In the anemic case (Figs 4a and 5a), the reduced viscosity and hematocrit yield a smooth, streamlined flow with well-aligned trajectories and minimal recirculation near branch points. The velocity vectors reinforce this, revealing uniform magnitudes and gradual directional changes. These features indicate low resistance and energy dissipation, reflecting the

ease of blood transport in low-viscosity conditions. Clinically, this may reduce arterial wall stress but can impair effective perfusion in distal microcirculation if velocity becomes insufficient to sustain pressure gradients [30].

The diabetic case (Figs 4b and 5b) presents markedly more complex flow dynamics. Streamlines exhibit pronounced curvature and bifurcation-induced deflection, particularly along the inner aortic arch and near the brachiocephalic and carotid branches, reflecting the combined effects of geometry and elevated blood viscosity. Vector plots reveal steep velocity gradients and localized high-velocity regions, indicating increased shear and flow non-uniformity. In addition, persistent blue-colored streamlines observed near the vessel wall indicate low-velocity helical and secondary flow structures, which are amplified by higher viscosity and increased viscous resistance, contributing to increased residence time and disturbed near-wall shear environments. These features contribute to increased flow resistance and non-uniform flow behavior. Such disturbed flow topologies have been associated with endothelial dysfunction, altered nitric oxide production, and initiation of atherogenic processes in diabetic conditions [31,32].

Healthy Cases 1 and 2 (Figs 4c, 4d, 5c, 5d) display intermediate patterns. Streamlines are generally smooth but reveal mild curvature at branch points due to natural geometric constraints. Velocity vectors remain balanced across the domain, indicating that blood transport is governed by geometry rather than pathological rheology. Consistent with the rheological parameters in Table 3 and the viscosity–shear-rate relationship shown in Fig 3, Healthy Case 1 exhibits a slightly higher effective viscosity than Case 2 over physiologically relevant shear rates, which manifests as modest differences in outlet velocity distributions. These features are consistent with stable aortic flow that supports uniform wall shear distribution and minimizes mechanical stress on the endothelium.

Fig 6 quantifies the maximum, minimum, and average outlet velocities across all four blood conditions. Diabetic blood generally exhibits higher maximum and average velocities at selected outlets, particularly O2, reflecting the increased pressure gradients required to overcome elevated viscosity. Anemic blood tends to show lower velocity magnitudes across most outlets, consistent with reduced hematocrit and lower viscous resistance. Healthy cases cluster between these extremes, with modest differences between Case 1 and Case 2 attributable to inter-individual rheological variability.

Minimum velocities show stronger contrasts. Reduced minimum velocities at O3 and O4 are observed for diabetic and healthy cases, indicating localized flow deceleration and intermittent recirculation, which are clinically relevant for shear-induced platelet activation and potential thrombus formation [11,26]. In contrast, the anemic case exhibits comparatively uniform minimum velocities across outlets, reflecting more consistent forward flow with minimal resistance.

Fig 7 presents helicity distributions in the aortic arch for four physiological conditions: (a) anemic, (b) diabetic, (c) healthy (Case 1), and (d) healthy (Case 2). Helicity, defined as the scalar product of local velocity and vorticity vectors, quantifies the extent of rotational flow and is a marker of helical motion in pulsatile arterial environments. In the anemic case, helicity is substantially diminished, consistent with reduced blood viscosity and weakened secondary

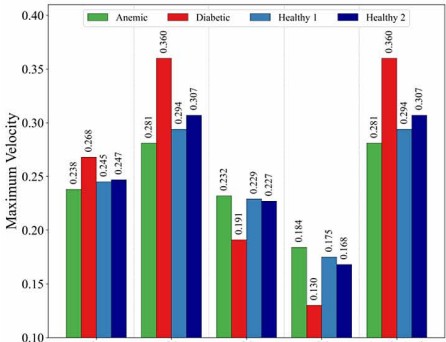 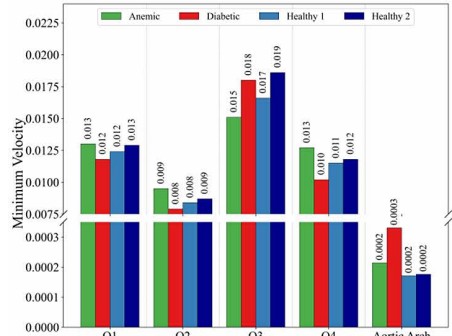 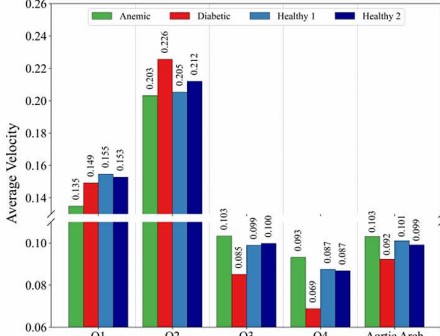

**Fig 6. Velocity distributions: average (left), minimum (middle), and maximum (right) across different outlets and cases.**

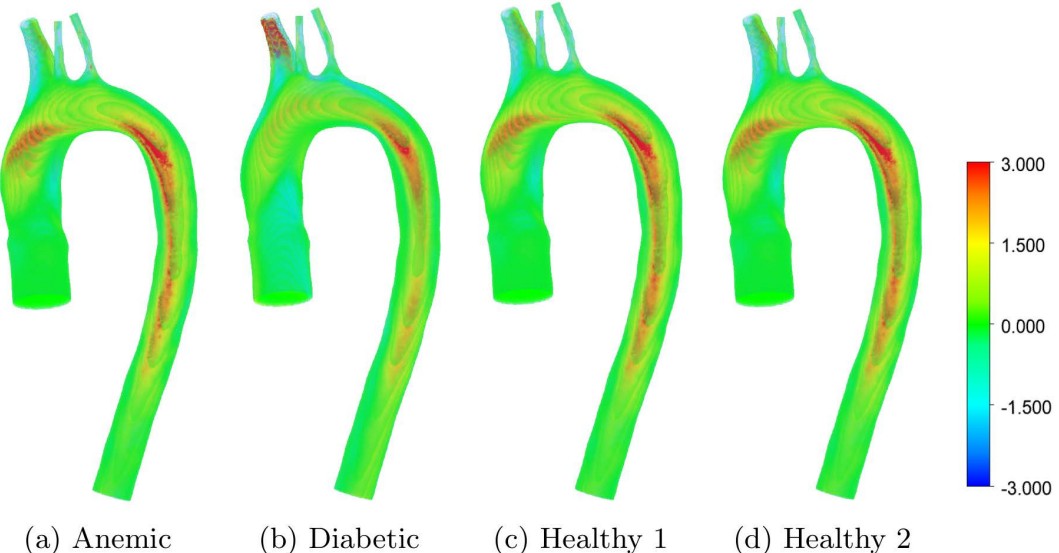

(a) Anemic      (b) Diabetic      (c) Healthy 1      (d) Healthy 2

**Fig 7. Helicity contours for four different patient-specific aortic flow cases: (a) anemic, (b) diabetic, (c) healthy (Case 1), and (d) healthy (Case 2).** The color scale indicates helicity values in $ms^{-2}$, with red and blue denoting strong positive and negative helicity regions, respectively.

flow structures. The diabetic model shows enhanced helicity, especially near the arch curvature and supra-aortic branches, suggesting intensified vortical dynamics likely associated with increased flow resistance and vascular stiffening. Both healthy cases demonstrate moderate helicity concentrated along the outer wall, reflecting physiologically favorable swirling patterns that enhance wall shear stress distribution and promote efficient transport of oxygen and nutrients. These findings underscore the role of helicity as a biomechanical indicator of vascular health and pathology in cardiovascular simulations.

Taken together, these results show how pathological changes in blood properties significantly alter velocity distributions, with implications for nutrient delivery, shear stress exposure, and cardiovascular risk. The combined use of streamlines, vectors, and velocity statistics provides a robust framework for characterizing aortic hemodynamics across physiological states. This multidimensional approach also enables identification of vulnerable flow patterns that may predispose to endothelial dysfunction or vascular remodeling under disease conditions.

### 3.2 Hemodynamic pressure distribution and flow resistance

Pressure distribution in the aortic arch provides critical insights into the flow resistance and hemodynamic forces acting on the arterial walls. Under pulsatile inlet conditions with RCR outlet coupling, the simulated pressure field reflects physiologically realistic wave propagation and damping across the arch. Abnormal pressure gradients are clinically associated with increased afterload, impaired organ perfusion, and progressive cardiovascular disease, particularly in patients with hypertension or systemic vascular stiffening [31].

Figs 8 and 9 illustrate the pressure contours for different blood conditions, comparing surface pressure (Fig 8) and volume-averaged pressure (Fig 9) distributions across four cases: anemic, diabetic, healthy (Case 1), and healthy (Case 2). The pressure patterns highlight the influence of blood viscosity, hematocrit levels, and vascular geometry on the overall flow resistance in the aortic arch.

In the anemic case (Figs 8a and 9a), the lower viscosity and hematocrit result in smoother flow with reduced pressure gradients across the aortic arch. The ascending aorta shows relatively uniform pressure, with minimal pressure drops near

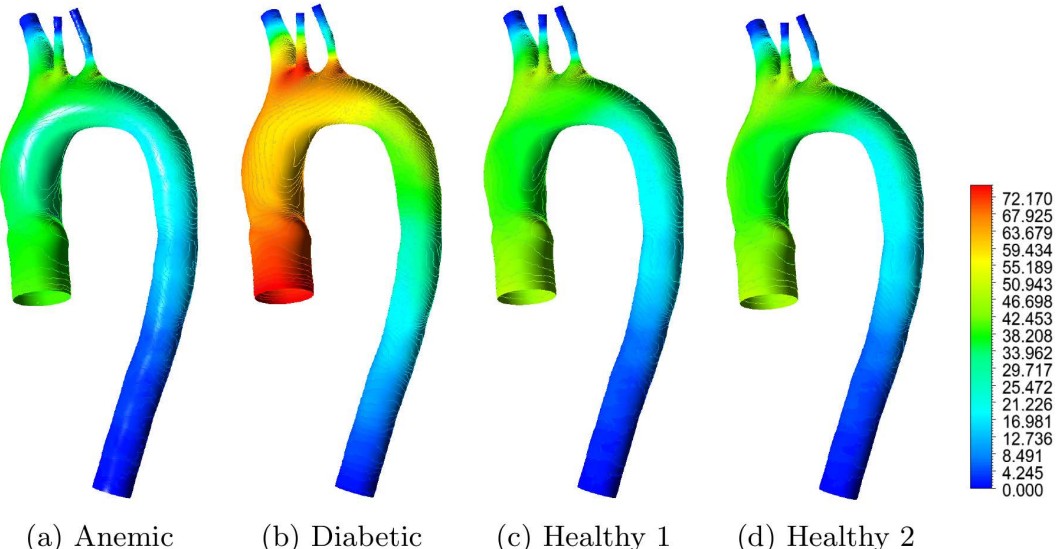

**Fig 8. Pressure contours in the aortic arch for different blood flow cases: (a) anemic, (b) diabetic, (c) healthy (Case 1), and (d) healthy (Case 2).** Red regions correspond to areas of higher pressure, while blue regions indicate lower pressure, with values presented in Pascal (Pa), highlighting variations due to different blood conditions and physiological factors.

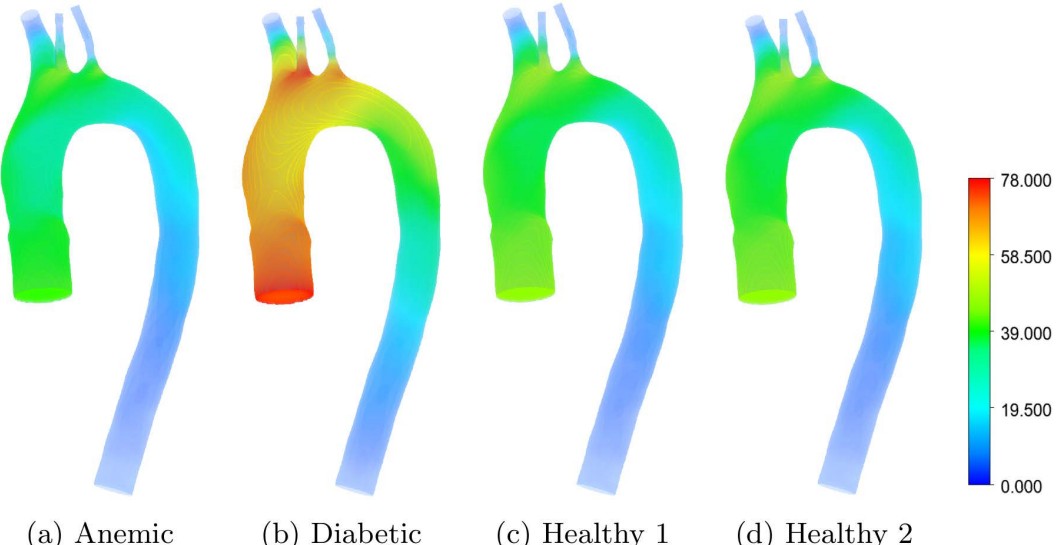

**Fig 9. Volume-averaged pressure contours in the aortic arch for different blood flow cases: (a) anemic, (b) diabetic, (c) healthy (Case 1), and (d) healthy (Case 2).** Red regions correspond to areas of higher pressure, while blue regions indicate lower pressure, with values presented in Pascal (Pa).

the outlets (brachiocephalic trunk, left carotid artery, and left subclavian artery). This reflects reduced resistance to flow and lower energy dissipation due to the lower viscosity of anemic blood. The descending aorta also displays smooth pressure variations, emphasizing the reduced overall flow resistance associated with this condition. Clinically, such low pressures

may be associated with reduced perfusion, especially in microcirculation, and may predispose patients to orthostatic intolerance or inadequate capillary oxygen delivery [27].

In contrast, the diabetic case (Figs 8b and 9b), in contrast, demonstrates significantly elevated pressure near the ascending aorta and along the inner curvature of the aortic arch. The higher viscosity and hematocrit of diabetic blood lead to amplified velocity gradients, increasing flow resistance and resulting in pronounced pressure drops near the branch outlets. The brachiocephalic trunk and left carotid artery exhibit the most substantial pressure differences, reflecting the greater energy loss associated with the more viscous fluid. The descending aorta also experiences higher pressure compared to the anemic case, highlighting the systemic effects of increased flow resistance in diabetic blood. These features are consistent with clinical observations of increased afterload and impaired pressure-buffering in diabetic vasculature [32].

Healthy cases (Figs 8c, 8d, 9c, and 9d) fall between the extremes of anemic and diabetic conditions. In Case 1, the pressure distribution near the ascending aorta and branch outlets is slightly higher than in Case 2, reflecting the moderate influence of hematocrit and viscosity. Both cases show localized high-pressure regions at the branch openings, indicative of flow acceleration and geometric effects, but these are more subdued compared to the diabetic case. The descending aorta in both healthy cases displays relatively smooth pressure variations, consistent with balanced hemodynamic conditions.

The comparison across cases underscores the role of blood rheology in shaping pressure distributions and flow resistance in the aortic arch. Anemic blood, with its lower viscosity, minimizes pressure drops and energy dissipation, leading to smoother flow patterns. Diabetic blood, on the other hand, imposes greater resistance, with higher pressure gradients and localized energy losses near bifurcations and outlets. Healthy cases strike a balance between these extremes, reflecting physiological conditions that maintain efficient blood flow while accommodating the natural geometric complexity of the aortic arch.

Fig 10a presents a quantitative comparison of inlet and wall pressures for all cases, based on maximum, average, and minimum values. Diabetic blood exhibits the highest pressures across all metrics, particularly at the wall, reflecting increased flow resistance due to elevated viscosity. In contrast, the anemic case consistently shows the lowest pressure values, which aligns with reduced hematocrit and energy dissipation. Healthy Cases 1 and 2 fall between these extremes, with Case 1 showing slightly higher pressures, possibly due to modest rheological differences. These pressure patterns corroborate the flow observations and provide a basis for evaluating systemic load and perfusion potential.

Fig 10b provides a categorical hemodynamic risk assessment intended as an illustrative tool based on minimum, maximum, and average wall pressure metrics. The classification logic is physiologically informed and intended to capture deviations from optimal hemodynamic conditions in a relative, comparative sense rather than as a clinically validated predictor. In pulsatile aortic blood flow, the minimum and maximum luminal pressures could be interpreted as approximations of diastolic and systolic pressures, respectively. Although CFD simulations typically do not model arterial wall compliance unless coupled with structural mechanics, these pressure extrema still offer useful hemodynamic proxies for comparative analysis when performing comparative risk analysis. The classification matrix emphasizes key hemodynamic contrasts across blood rheological states. The diabetic profile is flagged for elevated hemodynamic pressure severity, due to its elevated maximum wall pressure, consistent with increased vascular resistance and strain on the arterial wall. It also shows a borderline response in minimum and average pressure, suggesting incomplete normalization even during diastolic phases [31]. The anemic condition is marked by reduced pressure severity, particularly in the minimum metric, which reflects reduced circulatory force and potential for inadequate capillary perfusion. Both healthy cases, particularly Healthy 2 fall predominantly in the low-severity zone, reinforcing the interpretation that these conditions exhibit stable, well-regulated hemodynamics supportive of systemic vascular health.

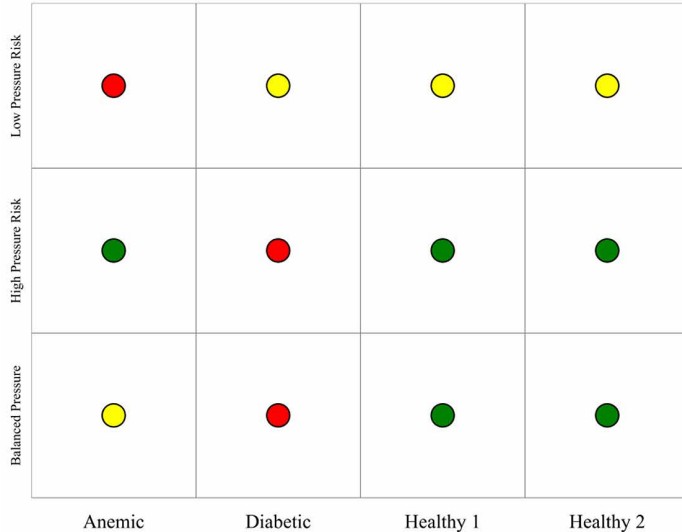

**Fig 10. (a) Pressure distribution across four blood conditions: anemic, diabetic, healthy (Case 1), and healthy (Case 2) showing average, maximum, and minimum values. Dotted bars indicate inlet pressure, while diagonally hatched bars represent wall pressure.** Diabetic cases exhibit elevated pressure magnitudes at both inlet and wall locations, while anemic cases show comparatively lower wall pressure values. **(b) Pressure-based hemodynamic risk classification (illustrative) using three metrics: minimum, maximum, and average pressure.** Color-coded circles represent relative hemodynamic severity levels, where red indicates high, yellow indicates borderline or moderate concern, and green indicates no risk.

## 3.3 Wall shear stress distribution and risk stratification

Wall shear stress (WSS) is a critical hemodynamic quantity representing the tangential force exerted by blood flow on vascular endothelium. It plays a central role in regulating endothelial function, vascular remodeling, and the development of atherosclerosis. Clinical and experimental studies have shown that deviations from physiological WSS values are associated with endothelial dysfunction, plaque formation, and thrombotic risk [2,33,34]. In particular, low WSS promotes leukocyte adhesion, platelet aggregation, and pro-inflammatory signaling, while abnormally high WSS may induce endothelial denudation and arterial remodeling [35].

Fig 11 shows the spatial distribution of WSS under four blood conditions. The anemic case exhibits smooth, uniform flow with moderate WSS magnitudes throughout the aortic arch. This behavior can be attributed to reduced blood viscosity, which directly lowers wall shear stress for a given velocity gradient, and to low hematocrit, which indirectly contributes by smoothing velocity profiles and reducing near-wall shear gradients. While such flow may reduce mechanical stress on the endothelium, persistently low WSS may impair shear-mediated signaling pathways, such as nitric oxide synthesis, leading to altered endothelial function [36].

In contrast, the diabetic case shows elevated and highly non-uniform WSS, especially near the brachiocephalic trunk and left carotid artery. This arises from increased viscosity and reduced arterial compliance, both of which intensify velocity gradients in complex flow regions. These observations align with clinical reports of endothelial stress and vascular remodeling in diabetes [26]. The healthy cases lie between these two extremes: both show moderate WSS values with localized elevations near outlets due to physiological flow acceleration and geometric curvature.

Fig 12a quantifies these observations. The diabetic case yields the highest maximum WSS (6.01 Pa), exceeding commonly reported physiological ranges, while the anemic case shows the lowest average (0.041 Pa) and minimum (0.005 Pa) WSS. Healthy cases remain mostly within the accepted physiological range of 0.4 to 1.5 Pa [2]. Panel b

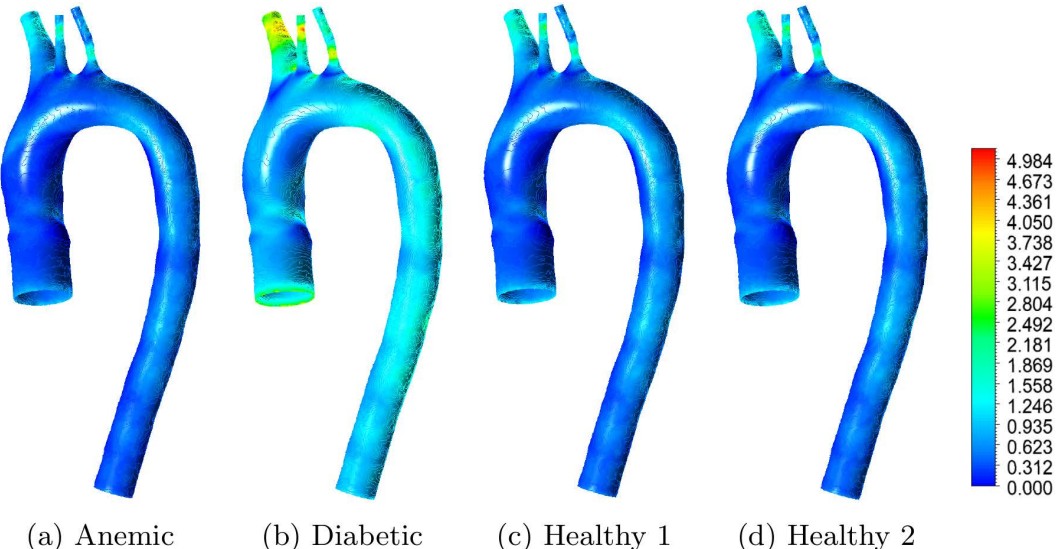

(a) Anemic     (b) Diabetic     (c) Healthy 1     (d) Healthy 2

**Fig 11. Wall shear stress (WSS) contours in the aortic arch for different blood flow cases: (a) anemic, (b) diabetic, (c) healthy (Case 1), and (d) healthy (Case 2).** Red regions correspond to higher WSS values, while blue regions indicate lower WSS values in Pascal (Pa).

translates these metrics into a color-coded illustrative hemodynamic severity map: the diabetic case shows elevated WSS severity, while the anemic case approaches the low-WSS threshold (yellow). The healthy cases are within the low-severity zone across all three metrics. These results confirm that blood rheology modulates the local hemodynamic environment, with important hemodynamic implications relevant to vascular health. Anemia leads to sub-threshold WSS that may impair mechanosensitive endothelial responses, whereas diabetes imposes excessive WSS variations that can cause mechanical endothelial stress. These findings reinforce the role of WSS as a hemodynamic indicator associated with vascular dysfunction and support its use as a comparative hemodynamic metric in mechanistic studies [32,35]. From a translational perspective, identifying regions of abnormal WSS can inform relative susceptibility for intimal thickening, aneurysmal remodeling, or thrombus initiation, particularly in high-risk groups such as diabetics and anemic individuals [30,37].

Overall, the comparative analysis across four representative blood conditions highlights the critical role of rheological variation in modulating aortic hemodynamics. Velocity fields, pressure distributions, and wall shear stress patterns each reveal distinct mechanistic signatures associated with anemic, diabetic, and healthy flows. These signatures not only reflect energy transport and flow efficiency but also indicate biomechanical environments that may promote endothelial dysfunction, thrombus formation, or vascular remodeling. The integration of simulation-derived hemodynamic metrics with established clinical risk factors could provide a non-invasive framework for stratifying cardiovascular risk and guiding personalized intervention strategies [31,37].

## 4 Conclusion

This study presented a comprehensive computational framework to examine blood flow dynamics in the aortic arch across four physiological and pathological cases: anemic, diabetic, and two healthy conditions. By employing the Carreau model for non-Newtonian viscosity, the simulations captured realistic blood rheology and enabled detailed analysis of velocity profiles, pressure fields, and wall shear stress distributions. The results underscore the importance of integrating blood properties, vessel geometry, and boundary conditions to understand vascular mechanics and their clinical consequences.

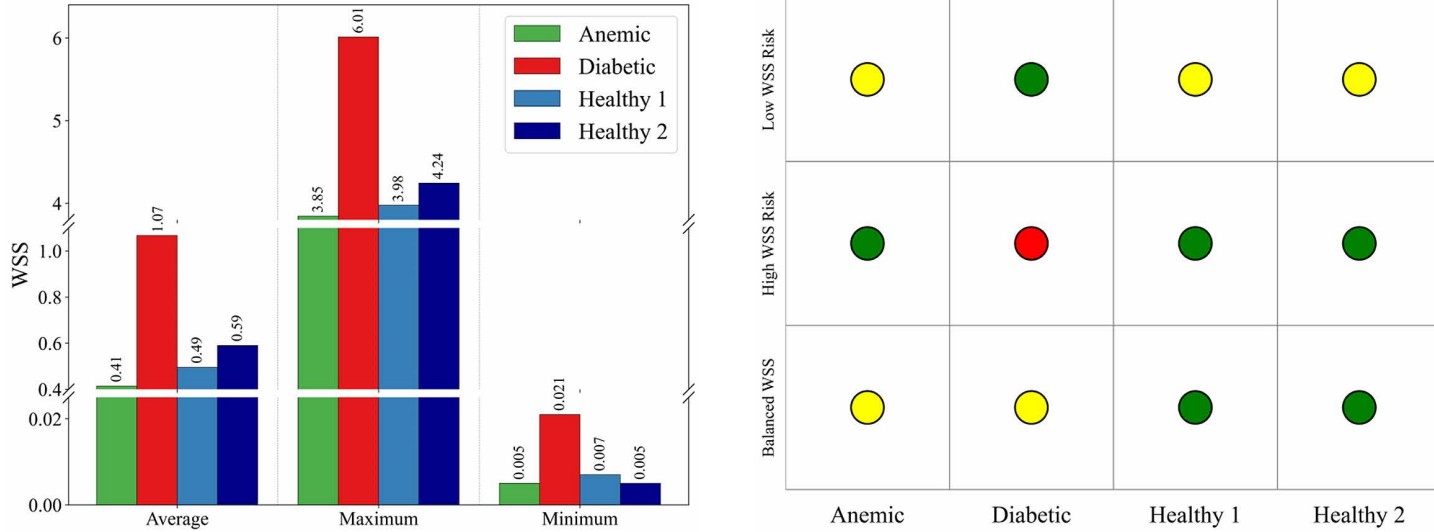

**Fig 12. (a) WSS distribution across four blood conditions–anemic, diabetic, healthy (Case 1), and healthy (Case 2)–showing average, maximum, and minimum values.** A broken y-axis is used to accommodate both low (below 0.05 Pa) and high (above 1 Pa) WSS values. **(b) Illustrative hemodynamic risk classification based on three WSS metrics: minimum, maximum, and average WSS.** Each cell is color-coded to represent the relative hemodynamic severity under a given metric, where red indicates high, yellow indicates borderline or moderate concern, and green indicates no risk.

Key findings show that anemic flow, characterized by low hematocrit and viscosity, resulted in smooth velocity streamlines, reduced pressure gradients, and uniformly low WSS values. Although energetically efficient, this condition may limit endothelial stimulation and impair perfusion, particularly in downstream or microvascular territories. Diabetic blood flow, in contrast, exhibited elevated viscosity and shear, which led to steeper velocity gradients, heightened WSS magnitudes, and significant pressure drops at outlet branches. These mechanical features are closely associated with pro-atherogenic stimuli, endothelial dysfunction, and increased cardiac workload, thereby validating the clinical association between abnormal blood rheology and vascular risk.

The two healthy cases demonstrated intermediate flow behavior, with velocity and WSS remaining largely within physiologic ranges. Flow acceleration near bifurcations and mild pressure asymmetries reflected normal anatomical constraints rather than pathology. Importantly, the study showed that hemodynamic indicators such as localized WSS peaks or recirculation zones can serve as proxies for disease-prone regions—even in geometries not yet anatomically compromised. These patterns reveal the systemic consequences of altered blood properties, offering mechanistic insight into how seemingly mild changes in rheology may initiate or amplify cardiovascular stress across different regions of the aorta.

Despite these insights, the study has certain limitations. The simulations were conducted on a generalized aortic geometry rather than patient-specific models, which may limit anatomical specificity. Blood flow was assumed to be laminar and incompressible, with rigid arterial walls and thus excluding potential contributions from fluid-structure interaction and wall elasticity. Additionally, while the Carreau model captures shear-thinning effects, it does not account for pathological viscoplasticity seen in conditions like thrombosis or inflammation. Moreover, shear-rate dependent viscosity models do not explicitly resolve time-dependent red blood cell aggregation, shear-history effects, or spatial variations in aggregation dynamics that may arise under pulsatile flow. In particular, axial and radial shear variations and local hematocrit redistribution were not modeled, and such spatiotemporal microstructural effects may be more pronounced in pathological blood where aggregation behavior differs. Experimental validation was beyond the scope of this work but remains essential to

confirm key predictions through experimental (e.g., particle image velocimetry) or clinical imaging (e.g., 4D Flow MRI) studies, particularly regarding shear stress thresholds and risk classification.

Future research should address these limitations by extending the present framework to patient-derived vascular geometries, incorporating compliant wall models through FSI, and expanding simulations to pathological cases such as stenosis and aneurysm. The present study is intentionally restricted to a non-diseased aortic geometry to establish a hemodynamic baseline, upon which pathological conditions can be systematically introduced. Incorporating alternative rheological models like Herschel–Bulkley or Casson or more comprehensive constitutive formulations such as thixo–elasto–visco–plastic (TEVP) models could improve accuracy in diseased states [38,39]. Coupling CFD with thrombotic and inflammatory pathways will enable richer physiological insights, while machine learning frameworks may enhance speed and predictive capability. Ultimately, this modeling approach can contribute to non-invasive vascular risk stratification and support the design of personalized therapeutic strategies.

## Author contributions

**Conceptualization:** Farzana Akter Tina, Hashnayne Ahmed.

**Data curation:** Farzana Akter Tina.

**Formal analysis:** Farzana Akter Tina, Hashnayne Ahmed.

**Investigation:** Farzana Akter Tina, Hashnayne Ahmed.

**Methodology:** Farzana Akter Tina, Hashnayne Ahmed.

**Resources:** Farzana Akter Tina.

**Software:** Farzana Akter Tina.

**Supervision:** Hashnayne Ahmed, Hena Rani Biswas.

**Validation:** Farzana Akter Tina, Hashnayne Ahmed.

**Visualization:** Farzana Akter Tina, Hashnayne Ahmed.

**Writing – original draft:** Farzana Akter Tina, Hashnayne Ahmed.

**Writing – review & editing:** Farzana Akter Tina, Hashnayne Ahmed, Hena Rani Biswas.

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
