## [Decision Letter · Decision Letter 0]

29 Oct 2025

Dear Dr. Ahmed,

Thank you for submitting your manuscript to PLOS ONE. After careful consideration, we feel that it has merit but does not fully meet PLOS ONE’s publication criteria as it currently stands. Therefore, we invite you to submit a revised version of the manuscript that addresses the points raised during the review process.

**ACADEMIC EDITOR:**

We look forward to receiving your revised manuscript.

Kind regards,

Dr. Hafiz Muhammad Umer Farooqi

Academic Editor

PLOS ONE

2. Please include a separate caption for each figure in your manuscript.

Additional Editor Comments:

Following a careful review of the manuscript and the referees’ constructive comments, it has been decided that substantial revisions are required prior to further consideration. The authors are encouraged to revise the manuscript thoroughly, addressing all reviewer comments and providing detailed explanations for the changes made.

Reviewers' comments:

Reviewer's Responses to Questions

**Comments to the Author**

1. Is the manuscript technically sound, and do the data support the conclusions?

Reviewer #1: Partly

Reviewer #2: Yes

2. Has the statistical analysis been performed appropriately and rigorously?

Reviewer #1: N/A

Reviewer #2: N/A

3. Have the authors made all data underlying the findings in their manuscript fully available?

Reviewer #1: No

Reviewer #2: Yes

4. Is the manuscript presented in an intelligible fashion and written in standard English?

Reviewer #1: Yes

Reviewer #2: Yes

Reviewer #1: This computational study seeks to investigate and compare the hemodynamic effects of anemic, diabetic, and healthy blood conditions within the human aortic arch. Its primary objective is to elucidate how pathological alterations in blood rheology—specifically viscosity and hematocrit—influence key hemodynamic parameters such as velocity, pressure, and wall shear stress (WSS), thereby providing insight into associated cardiovascular risks. The research addresses a notable gap in the literature by offering a direct, systematic comparison of these distinct blood states under a unified simulation framework.

While the study provides a valuable foundational analysis, several methodological aspects require refinement to meet the high standards for publication in a journal such as PLOS One. Addressing the following points would significantly strengthen the manuscript's contribution:

* The use of a constant static pressure at the outlets is a significant simplification. In vivo, the smaller arteries branching from the aorta present complex, dynamic resistance and compliance. This simplification likely affects the accuracy of flow distribution and pressure drops among the branches, which are critical for assessing regional hemodynamics.

* The aortic model is relatively smooth and assumes a non-diseased state. Consequently, the findings may not be generalizable to patients with common pre-existing conditions like atherosclerotic plaques, stenosis, or aneurysms, where geometric abnormalities dramatically alter local flow patterns and WSS.

* The study employs standardized parameters from the literature to define "anemic," "diabetic," and "healthy" blood. While based on real data, this approach treats each condition as a single, uniform state. In clinical reality, these conditions exist on a wide spectrum of severity, leading to a corresponding range of blood viscosities. A more nuanced analysis considering this variability would enhance the clinical relevance of the findings.

* The Carreau-Yasuda model effectively captures the shear-thinning behavior of blood but neglects other non-Newtonian characteristics. Recent advances in hemorheology, as detailed in works by Giannokostas and colleagues (e.g., J. Non-Newton. Fluid Mech., 2023; 2021; Materials, 2020), highlight the importance of properties like viscoelasticity, yield stress (viscoplasticity), and thixotropy. These properties are particularly critical in low-shear and recirculation zones, which are pivotal for understanding phenomena like thrombus formation. Incorporating these advanced models would substantially improve the physiological fidelity of the simulations. Furthermore, to accurately capture the complex hematological variations that distinguish anemic, diabetic, and healthy blood, a Thixo-Elasto-Visco-Plastic (TEVP) model is necessary. Such a model can uniquely account for the underlying rheological changes driven by variations in hematocrit, as well as the aggregability of red blood cells (RBCs) and platelets.

* The authors note that experimental validation was beyond the study's scope; however, this remains a critical step for establishing model credibility. Corroborating the results with experimental data (e.g., via particle image velocimetry) or clinical imaging (e.g., 4D Flow MRI) is essential to transition the predictions from theoretical to validated findings.

* The risk classification tables (Figures 9b and 11b) offer an effective visualization of the hemodynamic findings. It is important, however, to explicitly frame them as illustrative conceptual tools rather than validated clinical instruments. Clinical risk is a multifactorial continuum influenced by genetics, systemic inflammation, and lipid profiles, factors beyond the scope of this purely hemodynamic assessment.

Reviewer #2: This paper simulated the flow based on CFD in the human aortic arch for people, anemic, diabetic, and 2 healthy subjects with Carreau-Yasuda model of hemorheology under pulsatile flow. Even though this study combines hemodynamics and hemorheology with quantitative analysis, this requires extensive revision before publication.

1. Compared with previous published papers such as [7] & [20] in the reference, what is the originality of this paper? The authors should persuade this more rigorously.

2. The authors claimed that this CFD study is a direct comparison of pathological and physiological blood states for rheological interpretation under identical boundary and geometric conditions. However, it’s not clear how pathological blood would be reflected in the rheological effects in CFD. If authors meant that Carreau viscosity model with different coefficients according to pathology, are they good enough? Especially under pulsatile flow, red blood cell aggregation is not just a function of shear rate only but also a function of time and radial and axial positions depending on acceleration and hematocrit. Red cell aggregations are also dependent on axial shear rate in addition to radial shear rate under pulsatile flow. How these complicated red cell aggregation phenomena would be reflected in this simple model? Please state this with references.

3. Discussion needs to be separated from the results, and limitations need to be part of discussion rather than conclusion.

4. For governing equations, please clarify whether they are in cylindrical coordinate or cartesian coordinate including helicity eq. (5), For Eqs (3) & (4), n are not the same so different notation is required.

5. What kind of medical imaging data was used? More specific description is required in line 130.

6. In Fig. 2, (a) is not required, but (b) with more specific mesh and grid should be included in addition to overall mesh map.

7. Table 1 & 2, it not clear why 1mm of grid size and 0.4 s are optimal.

8. In addition to Table 3, the graphs are required to figure it out how viscosity would be changed with shear rate for 4 different blood samples.

9. In line 197 & 198, vessel geometry 197 shape overall flow characteristics should be rewritten.

10. In Fig. 3 and after, numbers of color bar should be replaced with real value rather than xe-01 for easier reading. Explain more details of the blue lines like helical flow close to wall.

11. In line 225, how are these suggested marginally higher viscosity and hematocrit?

12. Fig.5 & 9, the descriptions are not consistent with the bars in several expressions so please check them and rewrite them. How did you divide to three groups, red, greed, and yellow? Specify the references for each group.

13. In line 262, why did you set 0 Pa for the outlet pressure and this doesn’t seem to be physiological.

14. In 347&8, how would you justify this with low hematocrit and viscosity? Can you evaluate each of these effects from hematocrit and viscosity?

**Do you want your identity to be public for this peer review?** For information about this choice, including consent withdrawal, please see our Privacy Policy

Reviewer #1: No

Reviewer #2: **Yes:** Dong-Guk Paeng

---

## [Author Response · Author response to Decision Letter 1]

18 Jan 2026

We have provided a detailed, point-by-point response to all reviewer and editor comments in the attached “Response to Reviewers” document. All suggested revisions have been carefully addressed, and the manuscript has been updated accordingly. Both a marked-up version highlighting changes and a clean revised manuscript have been submitted for review.

---

## [Decision Letter · Decision Letter 1]

11 Feb 2026

Dear Dr. Ahmed,

Thank you for submitting your manuscript to PLOS ONE. After careful consideration, we feel that it has merit but does not fully meet PLOS ONE’s publication criteria as it currently stands. Therefore, we invite you to submit a revised version of the manuscript that addresses the points raised during the review process.

We look forward to receiving your revised manuscript.

Kind regards,

Hafiz Muhammad Umer Farooqi

Academic Editor

PLOS One

Journal Requirements:

Additional Editor Comments:

The manuscript has been significantly revised and improved. However, a few vital points still need to be addressed. Hence, the authors are advised to revise the manuscript accordingly and address all queries raised.

Reviewers' comments:

Reviewer's Responses to Questions

**Comments to the Author**

Reviewer #1: All comments have been addressed

Reviewer #2: (No Response)

2. Is the manuscript technically sound, and do the data support the conclusions?

Reviewer #1: Yes

Reviewer #2: Yes

3. Has the statistical analysis been performed appropriately and rigorously?

Reviewer #1: N/A

Reviewer #2: N/A

4. Have the authors made all data underlying the findings in their manuscript fully available?

Reviewer #1: Yes

Reviewer #2: Yes

5. Is the manuscript presented in an intelligible fashion and written in standard English?

Reviewer #1: Yes

Reviewer #2: Yes

Reviewer #1: I propose the acceptance of the manuscript for publishing. The authors have addressed all issues mentioned in my previous report.

Reviewer #2: Most of the review comments were satisfactorily answered. However, there are some more which should be addressed.

1. There are little differences of dynamic viscosity with shear rate between anemic blood and healthy blood 1. Rather there are more differences between two healthy bloods. Therefore, the results would be following those differences by the CFD. Please quantify the differences between 2 healthy subjects and anemic and healthy blood 1. Add the discussion this point too.

2. Authors described that anemic blood shows the lowest viscosity, but that’s true for only > 1 s-1. Low shear rate is more important for viscosity and blood flow. This should be carefully analyzed and discussed.

3. The color of 4 cases should be consistent for all figures for easy readibility.

4. The red blood cell aggregation and viscosity as a function of axial shear rate in addition to radial shear rate and local hematocrit under pulsatile flow are spatiotemporal variations and these are not discussed fully without any references. Even though this study could not integrate these spatiotemporal variations, it’s important to be discussed as a limitation. It may be more important for pathological blood between healthy and anemic or diabetic bloods since there are more possibilities that their red cell aggregation tendency would be different, which are not fully reflected by the viscosity function as a shear rate.

**Do you want your identity to be public for this peer review?** For information about this choice, including consent withdrawal, please see our Privacy Policy

Reviewer #1: No

Reviewer #2: **Yes:** Dong-Guk Paeng

---

## [Author Response · Author response to Decision Letter 2]

14 Feb 2026

Please check the attached "Response to Reviewers" file. Thanks.

---

## [Decision Letter · Decision Letter 2]

18 Feb 2026

Hemodynamic simulation in the aortic arch under anemic, diabetic, and healthy blood flow conditions using computational fluid dynamics

PONE-D-25-52209R2

Dear Dr. Ahmed,

We’re pleased to inform you that your manuscript has been judged scientifically suitable for publication and will be formally accepted for publication once it meets all outstanding technical requirements.

If your institution or institutions have a press office, please notify them about your upcoming paper to help maximize its impact. If they’ll be preparing press materials, please inform our press team as soon as possible – no later than 48 hours after receiving the formal acceptance. Your manuscript will remain under strict press embargo until 2 pm Eastern Time on the date of publication. For more information, please contact onepress@plos.org.

Kind regards,

Hafiz Muhammad Umer Farooqi

Academic Editor

PLOS One

Additional Editor Comments:

The authors have revised the manuscript in accordance with the reviewer’s suggestions and have provided satisfactory responses to the queries raised. It is hoped that the authors will further consider the reviewer’s suggestions regarding spatiotemporal red cell aggregation or local hematocrit under pulsatile flow and address these aspects in their future research.

After careful evaluation of the revised manuscript, it is concluded that the manuscript meets the scientific criteria required for publication in PLOS ONE. Therefore, the manuscript is accepted for publication.

Reviewers' comments:

Reviewer's Responses to Questions

**Comments to the Author**

Reviewer #2: All comments have been addressed

2. Is the manuscript technically sound, and do the data support the conclusions?

Reviewer #2: Yes

3. Has the statistical analysis been performed appropriately and rigorously?

Reviewer #2: N/A

4. Have the authors made all data underlying the findings in their manuscript fully available?

Reviewer #2: Yes

5. Is the manuscript presented in an intelligible fashion and written in standard English?

Reviewer #2: Yes

Reviewer #2: Every comment was addressed appropriately, except for the spatiotemporal red cell aggregation or local hematocrit under pulsatile flow. Hopefully, authors can understand the importance of these dynamic variations of red blood cell aggregation and viscosity under pulsatile flow.

**Do you want your identity to be public for this peer review?** For information about this choice, including consent withdrawal, please see our Privacy Policy

Reviewer #2: **Yes:** Dong-Guk Paeng

---

## [Editor Report · Acceptance letter]

PONE-D-25-52209R2

PLOS One

Dear Dr. Ahmed,

I'm pleased to inform you that your manuscript has been deemed suitable for publication in PLOS One. Congratulations! Your manuscript is now being handed over to our production team.

Kind regards,

on behalf of

Dr. Hafiz Muhammad Umer Farooqi

Academic Editor

PLOS One